# Dermatological Management of Aged Skin

Ewelina Rostkowska [1], Ewa Poleszak [2], Katarzyna Wojciechowska [2] and Katarzyna Dos Santos Szewczyk [3,*]

1   Student Research Group Belonging to Chair and Department of Applied Pharmacy, Medical University of Lublin, Chodzki 1, 20-093 Lublin, Poland
2   Chair and Department of Applied Pharmacy, Medical University of Lublin, Chodzki 1, 20-093 Lublin, Poland
3   Department of Pharmaceutical Botany, Medical University of Lublin, Chodzki 1, 20-093 Lublin, Poland
*   Correspondence: k.szewczyk@umlub.pl

**Abstract:** The subject of the work concerns the dermatological management of patients mainly with aged skin. The purpose of the work was to present the basic techniques and preparations which are performed by dermatologists in the treatment of aged skin. There are dermatological treatments related to the treatment of skin diseases and cosmetic treatments which are mainly related to skin care. In this work, the method of literature research was applied. On the basis of books and journal articles on dermatological and cosmetic procedures for aged skin, an analysis of treatment types was made. Then, the results of this analysis were presented in the paper under discussion. The paper presents information on the skin and its properties. The structure and functions of the skin, aging processes and characteristics of aged skin were discussed. Then, the possibilities of reducing the visible signs of skin aging through the use of invasive and non-invasive dermatological and cosmetological treatments were given, and the most important components of preparations used supportively in combating skin aging processes were discussed.

**Keywords:** aged skin; skin aging; aging processes factors; skin care methods; cosmetic ingredients; dermatological treatments

## 1. The Skin and Its Properties

### 1.1. Structure and Functions of the Skin

The skin, as the body's coating, performs many functions. It provides protection against external factors such as cold, heat, moisture, bacteria and chemicals. It is flexible and stretchy, making it possible for a person to move in complex ways. It transmits nerve impulses about phenomena occurring in the human environment [1]. Therefore, the skin is one of the basic elements of the human body, largely determining its health. Two basic functions of the skin are indicated—passive functions related to the protection of the body, and active functions, related to regulatory, sensory, defensive and secretory phenomena. The most important tasks performed by the skin within the framework of passive functions are [2]:

- protection of the body from the outside temperature,
- protection from the harmful range of solar radiation and moisture,
- protection from excessive or dynamic mechanical action—impacts, pressure, friction—as the skin is flexible and absorbs some of the energy,
- protection from chemicals,
- protection against microorganisms and biological agents.

The passive functions of the skin can also include the social function, relating to giving a person's external appearance and external personal characteristics, which is important in forming personal social relationships. On the other hand, the tasks included in the active function of the skin include [2]:

- regulating body temperature through various defense and adaptive mechanisms including the circulatory system,

- regulating the body's water and electrolyte metabolism, i.a., through excretory activities,
- participation in the defense mechanism against infections and pathogenic phenomena
- absorption (resorption) activities,
- participation in the metabolism of proteins, lipids, vitamins (synthesis of vitamin D3),
- conduction of sensory information, moisture, temperature, pressure.

Human skin consists of three main layers: epidermis, dermis and subcutaneous (adipose) tissue, within which are appendages (hair follicles, sebaceous and sweat glands, nails), as well as blood vessels and nerves. The outer layer—the epidermis—is composed of cells called keratinocytes, which are arranged in four layers [3,4]:

- stratum corneum—it can be described as the outermost layer of the epidermis. Its cells undergo continuous exfoliation. It consists of completely flattened, dead, as well as closely arranged cells without a nucleus (corneocytes). This layer contains compounds with hygroscopic properties, which are part of the NMF (natural moisturizing factor) such as amino acids, pyroglutamic acid (PGA) and its sodium salt, urea, uric acid and glycosamine,
- granular—this is located just below the stratum corneum and consists of several levels of individual flattened cells; in this part are formed the keratin knots responsible for the color of the skin,
- squamous—composed of several levels of strongly interconnected cells, this is where ceramides are produced, which are substances that in the layers of keratinized epidermis form the cohesiveness of the skin,
- basal, also called proliferative—this is a row of single cells, i.e., keratinocytes, which sit directly on the basement membrane. This is a thin zone where the division of the basal cells of the skin and their growth towards higher layers takes place. In the newly formed cells there are also specialized cells responsible for the immune system (Langerhans cells), protecting the skin (melanocytes) and cells working as receptors.

The dermis is about 1–3 mm thick and determines the elasticity and resilience of the entire skin structure. It consists mainly of protein in the form of collagen (75%) and elastin (2–4%). This layer of skin also contains hygroscopic substances, for example hyaluronic acid, that regulate the amount of water in the skin. Proteoglycans provide hydration and viscosity to the dermis, while elastin is responsible for normal flexibility and elasticity. The dermis is composed of approximately 60–70% water [5,6]. As we age, human skin contains less and less water and sags [2].

The dermis also contains sebaceous glands and sweat glands. Sebaceous glands are distributed throughout the body, and although they perform a similar task, they vary in size and shape in different zones of the skin. They secrete sebum, which on the surface of the skin acts as a preservative and lubricant and protects against biological infections [5].

The last layer of skin is subcutaneous tissue. It consists of intertwined, compact fibers of connective tissue and flakes of fat. Blood vessels that supply the skin also run through this layer. The subcutaneous layer has thermal insulating properties and protects internal organs from injury [5].

The general structure of male and female skin is identical. However, there are some significant differences in terms of its individual components. First of all, men's skin is thicker, by about 20–25%, in relation to women's skin. These differences are mainly due to the thicker stratum corneum in men, which makes men's skin rougher and harder to the touch than women's skin. Nerve stimuli such as heat and mechanical pressure are less well conducted by the nerves, but the thicker epidermis and stratum corneum better protect the skin from mechanical damage and injury [6].

Women's wrinkles are flatter than men's, but they appear earlier. Men have deeper wrinkles, but their formation process is later than in women's. Male skin has a better blood supply, and a large number of hairs and adjacent sebaceous glands make the skin oilier, which can contribute to acne lesions. Some studies [6] indicate higher levels of sebum in men, which is related to sex hormones. However, there is emerging research indicating that women's skin can also show high levels of sebum. Among men, skin pigmentation is much

higher and there is more noticeable facial sagging in the lower eyelids compared to women. Skin elasticity among men and women does not differ significantly [6–10]. It is worth noting that the ways and habits of skin care are influenced by culture. An example is the Korean culture, in which men show more willingness to take care of their skin compared to other cultures [10–12].

There are differences between hairy skin and glabrous skin. One of these differences includes the aspect of touch. Glabrous skin does not have C-tactile (CT) afferents and is associated more with discriminative touch, whereas hairy skin has C-tactile (CT) afferents and is associated more with an affective touch [12,13]. The differences between these skin types are also related to differences in the stratum corneum thickness, the presence or absence of pilosebaceous units and also the perception [13].

### 1.2. Human Skin Condition and Aging

The skin is an integral part of the human body and, like the whole body, has a unique, genetically determined character. Aging processes result from both internal factors, directly related to a person and his specific organism (endogenous) and external factors (exogenous), i.e., the environment in which a person lives. The phenomenon of aging involves the successive weakening of the performance of the body's cells, which function less well in terms of metabolic and self-regulation processes and regenerative capacity. Both external and internal factors have a significant impact on the skin aging phenomenon, and there are various interactions between them that increase the effect of a particular factor [14].

The endogenous factors that influence the skin aging process are mainly [15,16]:

- natural and genetic predispositions related to aging,
- age—the biggest changes in women occur during menopause,
- general state of health, past skin diseases.

Around the time of menopause, which averages at around 50–52 years of a woman's life, the production of estrogen, a sex hormone that is also a regulator of collagen synthesis—the substance responsible for skin elasticity—decreases [17]. The loss of collagen in the skin is as high as 30% during the first 5 years after the onset of menopause, and in later years the loss is about 1–2% per year [18].

In addition, the amount of collagen fibers decreases, they have a different chemical composition and are corrugated, stiff and irregularly distributed, resulting in a decrease in skin thickness and loss of elasticity and tension. Due to lower amounts of estrogen, the ability to create new cells in the basal layer of the epidermis is also reduced. Because of this, the stratum spinosum and granular layer become thinner. The sebaceous and sweat glands are also less active during menopause, making the skin dry and rougher to the touch [19].

External factors that affect skin aging include [20,21]:

- improper diet, poor in antioxidants,
- exposure of the skin to ultraviolet (UV) radiation,
- exposure to various chemicals on the skin, associated with being in a contaminated environment and also passive and active smoking,
- improperly implemented facial care procedures,
- skin diseases—bacterial and viral, the presence of parasites in the skin, etc.

### 1.2.1. UV Radiation

One of the important external factors that has a major impact on skin aging is UV radiation, which has the strongest effect on the face and hands. The term "photoaging" describes the many destructive changes in the appearance, function and structure of the skin caused by excessive and prolonged exposure to UV radiation and artificial UV radiation, such as in tanning beds. UVA radiation (wavelength 320–400 nm) and UVB radiation (wavelength 290–320 nm) are responsible for photoaging. UV radiation penetrates the skin and leads to harmful effects, which depend on its intensity, wavelength and frequency of exposure, as well as the occurrence of sunburn. The degree of skin reaction to UV radiation

depends on the complexion of the skin (skin phototype). Those most prone to skin damage and subsequent aging are those with class I and II phototypes [22]. However, there is also evidence that for phototypes III and IV, photoaging may be a cause of skin aging. These skin types also present a greater predisposition to melanocytic changes, such as melasma and lentigo [23,24].

It is very important that photoaging of the skin is defined as the aging process that occurs in this tissue. The process means that any changes may be noticeable and observable before clinical symptoms appear. This can be used to increase better photoprotection [25].

Sunlight increases photoallergies and contributes to the formation of so-called free radicals responsible for skin cell aging. Reactions of reactive oxygen species (ROS) with proteins, lipid-protein membranes and nucleic acids are particularly dangerous. Smoking also contributes to the generation of free radicals and faster skin aging. The effect of free radicals is cell damage and destruction of its components. The effects of photoaging include deep furrows on the skin, nodules and papules. There are also hypertrophic changes, dryness and roughness of the skin, solar keratosis, deep wrinkles, and telangiectasias. Harmful effects of UV radiation can manifest themselves at different times in life, even years later, for example, in the form of cancerous lesions [26].

1.2.2. Blue Light

In addition to UV radiation, a serious threat to the skin is also a part of visible radiation, so-called blue light (BL), with a range of wavelengths (~400 nm–480 nm). The spectrum of blue light borders on UVA radiation. It is a heavy energy visible light (HEV) that has an effect on the skin similar (albeit 4x weaker) to UVA radiation. Shorter wavelength and thus higher energy light has a more destructive effect on the skin [27,28].

Both UVA and BL radiation cause increased hyperpigmentation of the skin, the formation of spots and hyperpigmentation which results from disrupted melanocytes [27]. In epithelial cells, BL causes the destruction of mitochondrial DNA and increases the production of oxygen free radicals and nitrogen free radicals. Increased oxidative stress in fibroblasts interferes with their division, resulting in reduced collagen and elastin synthesis and thus premature skin aging [27,29].

HEV rays penetrate deep into the body's tissues and lead to the activation of free radicals. This process results in the destruction of collagen and elastin fibers, which are responsible for skin firmness and elasticity. In addition, free radicals cause the destruction of skin DNA and lead to the elimination of lipids, or fat cells, which have a very important role in the issue of skin hydration. In a situation of insufficient lipids, the aging process becomes faster, and the skin becomes dull, dry and is stripped of its natural color. Facial hyperpigmentation can also appear, and for this reason people who have a particular tendency toward pigmentary changes should watch out for HEV radiation. Frequent and prolonged exposure to HEV radiation can result in an increase in the intensity of irritation and exacerbation of lesions, so people with sensitive skin should also be careful. In a situation of regular and frequent use of a phone or computer, care should be taken to protect the skin accordingly [30].

The skin's circadian rhythm can be disrupted by excessive BL in the evenings. Skin cells soothe inflammation and repair damage at night. However, the action of BL can cause cells to 'confuse' themselves and the natural regeneration process can be disrupted [31].

The skin can be protected from the effects of BL in various ways. Some of these may be, for example, limiting the amount of time that one spends in contact with a laptop or phone; being able to use such devices that are equipped with a BL filter; using cosmetics that have a high level of antioxidant potential; and paying attention to ensuring that the diet is rich in antioxidants, as they are involved in the process of destroying free radicals [31].

Protecting the skin from digital aging primarily involves using cosmetics rich in antioxidants like carotenoids, niacinamide, vitamin E and vitamin C and its derivatives [32]. Vitamin C is a very important antioxidant. This vitamin helps neutralize free radicals, smoothens the complexion and adds natural vitality. It also participates in the process of

lightening discolorations on the skin and provides protection against UV radiation. Vitamin E leads to the process of accelerating the restoration of cells and protects them from the negative effects of oxidative stress. Niacinamide is a very important ingredient in the process of skin regeneration, as well as in the process of combating the resulting discoloration [30].

### 1.2.3. Urbanization Aging

The growth of urbanization as well as the associated increased levels of environmental pollutants are noticeable in many regions of the world [33]. Air pollutants may include [34–36]:

- gases such as sulfur dioxide, carbon monoxide, nitrogen oxide;
- particulate matter (PM) which is characterized by varying particle sizes, and which may contain substances such as carcinogenic aromatic hydrocarbons (PAHs) or volatile organic compounds (VOCs);
- ozone which is formed by the reaction of pollutants with oxygen, with the involvement of UV action.

Sources of environmental pollution can include coal-fired power plants, car exhaust, cigarette smoke, and furnaces found in households. There is a growing awareness among the population of these phenomena and concern about the dangers that may result from them. Studies indicate that the mentioned factors have negative effects on human health. In addition, these factors have a high reactivity towards biological structures, including the condition of the skin. Air pollutants have effects related to a mechanism that is linked to their activity toward the acryl hydrocarbon receptor (AHR). Several types of skin cells characterize this system, for example, melanocytes, keratinocytes, fibroblasts and Langerhans cells. External environmental factors lead to its activation, and this leads to the expression of genes controlling reactions that are related to oxidative stress, immunosuppression, pigmentation induction, inflammation or premature skin aging [33].

Simple surface interactions of pollutants have the ability to lead to large changes in the natural composition of the outer structures of the skin layers. There is an increase in the amount of sebum secreted by the sebaceous glands, as well as a change in its composition in a negative way. The content of very important components of the lipid layer of the skin decreases: phospholipids, glucosylceramides, cholesteryl sulfate and sphingomyelins. There is also an increase in the amount of lactic acid formed and this accounts for the decrease in the skin's pH [33].

In addition, environmental pollutants contribute to the reduction of antioxidants, for example squalene or vitamin E, which are activated to fight oxidative stress, and which provide protection for the outer layers of the skin [33]. This process leads to the release of free radicals, which lead to the destruction of hyaluronic acid and the degradation of structural proteins, namely collagen and keratin, which provide elasticity to the skin and protect against water loss. Environmental pollutants contribute to damage to the natural protective barrier, deterioration of the cohesion of the stratum corneum, excessive skin sensitivity, and increased levels of the erythematous index. Pollutants lead to oxidative stress, which is a cause of faster aging but can also worsen other existing dermatological problems, such as psoriasis, acne, atopic inflammation, or even account for the appearance of skin cancerous lesions [33,35,37].

Cosmetics are increasingly appearing to reduce the negative effects of environmental pollution. These cosmetics are mainly related to skin care, which helps nullify the negative effects of pollution. They are, for example [38,39]:

- cleansing cosmetics, whose task is to remove the impurities on the skin;
- antioxidants that replenish those reduced by environmental pollutants and support the fight against antioxidant stress. This is mainly vitamin C and vitamin E;
- protective cosmetics, which have the effect of protecting against the penetration of pollutants;
- emollients, whose function is to maintain the integrity of the epidermal barrier and replenish it, such as ceramides, naturally occurring fatty acids among lipids and cholesterol.

### 1.3. Features of Aged Skin

The first sign of aged skin is wrinkles, which can be deep or superficial (up to 0.05 mm deep). As we age, wrinkles become more numerous and deeper, which is the result of a decrease in skin elasticity and loss of fat, water and thus a decrease in the thickness of the various layers [22]. One of the reasons for less elasticity of the skin after the age of 40 is the body's cessation of production of ceramides, components of the cellular cement, which are responsible for proper skin hydration. Atrophy of adipose tissue, most noticeable in the periorbital, temporal and suborbital regions, is responsible for the skin laxity of aging skin [40].

Skin aging involves all layers of the skin. The skin's ability to renew and regenerate is diminished. Restrictions occur in the metabolic exchange between skin layers, and the skin's immune defenses are impaired, making it more susceptible to infection. As the skin ages, it becomes drier, less smooth and more sensitive to external factors. In aging skin, elastic fibers become thinner and may even atrophy in the papillary layer so that collagen is unable to perform its functions. There is also atrophy of vertical vascular loops, resulting in reduced blood flow, reduced nutrient supply to the skin, impaired thermoregulation, reduced skin surface temperature and pale skin [41].

## 2. Opportunities to Reduce Skin Aging Processes in the Beauty Salon

### 2.1. Invasive Methods

The most commonly used invasive skin care methods include:

- platelet-rich plasma (PRP) treatment,
- needle mesotherapy,
- PDO (polydioxanone) lifting threads,
- botulinum toxin type A injection,
- carboxytherapy.

It must be noted that invasive procedures, such as breaking the continuity of the skin, must be performed by qualified individuals, such as a doctor. It is important to ensure the safety and effectiveness of the procedure and to minimize the risk of complications. Only a qualified doctor has the appropriate knowledge and skills to perform such a procedure.

### 2.1.1. Platelet-Rich Plasma (PRP) Treatment

Human blood is made up of plasma and the formed elements, or morphotic elements, which include erythrocytes (red blood cells), leukocytes (white blood cells) and thrombocytes. The most important function of platelet cells is to support homeostasis, that is, to stop bleeding on their own. This function prevents the body from bleeding out and keeps the blood in the placenta fluid. During treatment with platelet-rich plasma (PRP), plasma is injected into the human skin, which is enriched with platelets. This results in tissue reconstruction at the cellular level. The activator for the platelets is the first contact with collagen, the execution of the injection—that is, the trauma itself—chemicals such as calcium chloride, the heating of the body's tissues to an appropriate temperature as well as UV radiation [42].

PRP is properly prepared for the procedure by a blood centrifugation process. Once injected under the skin, the process of degranulation of alpha granules in platelets begins in the tissues. They secrete a growth factor that stimulates skin cells to regenerate. After the thrombocytes die off, macrophages take over the repair function. Preparing the preparation begins with the collection of about 10–20 mL of blood from the client into a syringe containing an anticoagulant in the form of dextrose citrate or sodium citrate. The collected blood is centrifuged in an angle or horizontal centrifuge. This obtains plasma separated from morphotic elements. The plasma is separated from the morphotic elements and placed in a syringe and then the injection is made into selected points of the skin. For full effect, it is advisable to perform a series of treatments, preferably three with intervals of one month. Treatment with platelet-rich plasma does not alter facial features and gives natural, incremental results. Because the client's blood components are used for the procedure,

the risk of allergies and infections is reduced [43]. The platelet-rich plasma treatment has rejuvenating and regenerative properties. It restores skin firmness, elasticity and is used in the treatment of acne scars and accelerates hair growth. A double-blind placebo-controlled randomized trial showed that PRP was able to significantly improve hair regrowth in alopecia areata. Results were significantly better after PRP was used than with placebo and steroid cream [44].

### 2.1.2. Needle Mesotherapy

An invasive procedure that is very commonly performed in beauty salons is needle mesotherapy. It is performed with a special device or traditionally with a needle [45]. An example of a needle mesotherapy device is shown in Figure 1 [46].

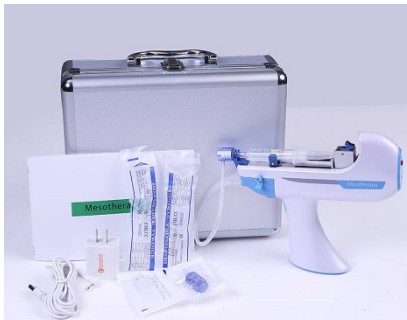

**Figure 1.** View of the OP-N needle mesotherapy gun [46].

This method uses various preparations containing hyaluronic acid alone or combined with a vitamins, inorganic salts, glycerin or trace elements. The product, appropriately matched to the patient's needs, is injected into previously anesthetized skin. The preparation is administered intradermally or subcutaneously. By interrupting the continuity of the epidermis, the healing and reconstruction processes of the skin are accelerated. The use of this method inhibits the loss of collagen, elastin, proteoglycans, as well as Langerhans cells. The main benefits of using needle mesotherapy are [47]:

- application of active substances to the dermis,
- nourishing, hydrating and rehydrating the skin,
- improving blood circulation and color,
- restructuring of the skin around the eyes,
- thickening and oxygenation of the skin,
- leveling existing wrinkles,
- preventing skin aging and sagging, on the abdomen, thighs and arms,
- contouring of the face.

With needle mesotherapy treatment, the skin is brighter, more hydrated, which allows for a lifting effect. For aged skin, two series consisting of 4–6 treatments per year are recommended [48].

### 2.1.3. PDO (Polydioxanone) Lifting Threads

Treatment with lifting threads is often used in aesthetic dermatology. The threads are made of polydioxanone, which is also used to produce dissolvable sutures in medicine. The procedure is performed under local anesthesia. Using cannulas or needles, PDO threads are implanted into the subcutaneous tissue [49]. In the case of tightening threads, a "scaffold" is created, so to speak, to soften the wrinkles that have formed. Lifting threads pull up the oval of the face, giving an anti-gravity effect. For a better effect, the thickness and length of the threads can be selected depending on the size of the face and the thickness of the client's skin. Most commonly, 1–3 PDO threads per side are used when using lifting threads. When using tightening threads, several to dozens of threads per area can be used [50]. The effects of using lifting threads can be seen as early as three weeks after the procedure. Significant

improvement is seen after another two to three months. The complete result of thread absorption occurs one year after the procedure [51].

### 2.1.4. Botulinum Toxin

The next type of invasive treatment is botulinum toxin type A injection. This is a safe and effective method of reducing facial wrinkles. Botulinum is one of the most popular preparations used today in medical and cosmetology offices. Neuromodulators were first used in medicine in 1950 in neurology and later in ophthalmology. In 2002, botulinum toxin was approved by the Food and Drug Administration (FDA) for the temporary improvement of moderate to deep wrinkles. It is used to smooth and shallow wrinkles, as well as to treat hyperhidrosis or bruxism. Botulism injection causes inhibition of the secretion of acetylcholine, the neurotransmitter responsible for the conduction of nerve impulses to muscles. Using a thin 30 G needle, the preparation is injected into the specific muscle causing the wrinkle, and as a result, relaxing the muscle and reducing the wrinkle. The effect lasts approximately 4–6 months [52].

Botulinum toxin type A, as well as B, is most commonly used for therapeutic purposes these days. Examples of preparations that include botulinum toxin type A, which are registered in Europe, are the following products with their trade names: Botox®, Dysport® and Xeomin®. In the United States, there is a registered preparation that has botulinum toxin type B in its composition, and this is a product whose trade name is MYOBLIC®; however, the product does not currently have approval for cosmetic use [53]. Nowadays, interest in the botulinum toxin is growing, and one can also see an increase in the number of indications for its use. Botulinum toxin has become popular mainly for its anti-wrinkle effect, which is associated with its wide use in the field of aesthetic medicine. Gradually, its range of therapeutic recommendations and pharmacological actions is expanding. At first, botulinum toxin type A received FDA approval for the temporary correction of deep forehead wrinkles. Allergan's therapeutic preparation Botox® was used for facial wrinkles in the upper part of the face. Studies have shown that botulinum toxin treatments are safe, as well as producing the desired results in terms of correcting facial wrinkles. Usually within 24–72 h after the procedure, the effect is already visible. Patients who are under 50 years of age achieve the best results with this type of treatment [54]. The development of research in this field, the expansion of knowledge, as well as the experience gained, has led to the fact that the use of the toxin has also been extended to the neck, décolleté and lower face [55]. A preparation containing botulinum toxin is injected into the relevant muscles, the dose of which is adjusted on an individual basis. Patients are advised to pay attention to rest after the procedure; it is advisable not to engage in high-exercise activities for a minimum of two days, and should not undergo facial massages and treatments or laser treatments for two weeks after the procedure [56]. Neuromuscular disorders, which may include, for example, pseudomuscular myasthenia gravis, as well as allergy to the toxin, are primary contraindications to botulinum toxin treatment [57].

The mechanism of action associated with the administration of botulinum toxin is that, once injected into the appropriate tissue, it binds using a heavy chain to the presynaptic terminals of cholinergic nerve fibers in the motor plate. The high affinity of receptors located on the surface of the presynaptic membrane with toxin molecules ensures this connectivity. Botulinum toxin type A is assigned a receptor, which is a protein fused to the membrane of the presynaptic vesicle SV2C [58]. When acetylcholine is released from the vesicle into the synaptic gap, it becomes possible for the SV2C protein to bind to botulinum toxin type A. The reason for this is that the SV2C receptor is located on the inner part of the presynaptic vesicle's membrane surface. The intensity of the toxin coupling process depends on the number of vesicles that release acetylcholine. The intensity increases if the number of acetylcholine-releasing vesicles increases. The intensity of the toxin binding process is greater if the neuromuscular synapse is more active. Botulinum toxin binds to the SV2C receptor and endocytosis occurs. The light chain characterized by enzymatic properties penetrates into the cytosol due to the interaction of the endosomal vesicle

membrane with the hydrophobic domains of the heavy chain of the toxin. The SNAP-25 protein in the cytoplasm is a substrate of the light chain, is part of the SNARE protein complex and is responsible for the transport and secretion of vesicles with acetylcholine into the synaptic space. The functions of SNARE proteins are related to the process of recognition and fusion of the cell membrane with vesicles and the process of vesicular transport. Among the SNARE proteins is SNAP-25, a synaptosome-associated protein; there is also synaptobrevin, a synaptic vesicle protein, and the cell membrane-associated syntaxin. Botulinum toxin type A and its effects are linked to the process of enzymatic damage in the presynaptic membrane of nerve-end proteins of the SNARE complex. The toxin's light chain is responsible for the defragmentation of the SNAP-25 protein [53]. The toxin is the cause of hydrolysis of the SNAP-25 protein, which leads to the process of cutting off the last nine amino acids; this in turn results in the fact that the release of acetylcholine from the presynaptic nerve fiber endings of the CNS is inhibited. Muscles undergo a process of relaxation and the function of the motor neuromuscular plate is blocked. Changes in the autonomic nervous system can also occur, for example, such as paralysis of eye accommodation, dryness of the mucous membrane of the throat and mouth and orthostatic drops in blood pressure [59].

### 2.1.5. Carboxytherapy

Carboxytherapy is the use of purified carbon dioxide in a controlled manner in various parts of the body. Such a treatment is used to improve skin elasticity and circulation in the tissues, as well as to reduce local excess fatty tissue or to improve the cosmetic effect after liposuction [60]. After application, carbon dioxide is removed from the human body as a product of gas exchange, that is, naturally. The amount of carbon dioxide applied in carboxytherapy is comparable to the amount of gas that is produced at the time of exercise [61].

Subcutaneous injection is related to the injection of $CO_2$ into the subcutaneous tissue to reduce cellulite, reduce local excesses in adipose tissue and improve blood circulation. Subcutaneous carbon dioxide injection can be used in the thighs, neck area, abdomen, supraspinatus, in the ileal area or in the back of the arm. The angle at which the needle is inserted is 45 degrees. Intradermal injection is associated with the injection of carbon dioxide to reduce stretch marks, improve the elasticity of the skin of the neck and face, heal or restore the eye area, improve the skin of the hands, as well as in situations of baldness. This type of injection can be used around the upper and lower extremities, torso and head. Injections in the area of the head require special care because there is a disruption of tissues at the site of the course of many blood vessels and nerves [62]. Carboxytherapy is associated with the application of gas using a thin needle, so it is a procedure characterized by the fact that it is minimally invasive. During this treatment, devices that have the function of regulating the pressure, flow and temperature of the applied gas should be used. Perceived discomfort during the procedure may be associated with a burning, pressure, spreading feeling, and the cause of such a condition is the injection and accumulation of gas. After the procedure, the patient can immediately begin daily activities, though swelling and mild redness may temporarily persist [63]. Contraindications that are most commonly mentioned for carboxytherapy procedure are heart disease, pregnancy and severe lung and kidney failure [64]. Carboxytherapy is a $CO_2$ therapy that is described as an effective and safe method that results in skin rejuvenation. This treatment is an alternative to methods characterized by greater invasiveness, such as surgical lifting and laser therapies [65]. Fat reduction is the main indication for carboxytherapy. This treatment can be the main factor in the process of reducing local excess fatty tissue, or it can be a complementary factor to the liposuction procedure, the aim of which would be to increase the intended results and make the skin smoother. Carboxytherapy is a method that has a mechanical and biochemical effect. For this reason, it is used for the purposes of correcting the figure, treating skin that is not very firm on the arms, abdomen and inner thighs, as well as correcting the neck, décolleté and double chin. Carboxytherapy is very effective in the treatment of scars and

stretch marks. This treatment stimulates the synthesis of collagen and elastin, leads to an improvement in blood supply to the tissues and results in the loosening of adhesions located in the subcutaneous tissue area. $CO_2$ therapy stimulates angiogenesis, and thus has an effect on fibroblasts, which results in the production of collagen. The new collagen gives the skin a smoothing effect, improving its appearance [66]. Carboxytherapy is also one of the most effective methods used in the process of reducing cellulite. This treatment affects the factors that are the cause of cellulite. Carbon dioxide injection is associated with an increase in lipolysis and damage to adipocytes, even in cases at an advanced level. The fibrotic structure is loosened, lymphatic flow is improved, and capillary flows are increased [67]. Carbon dioxide therapy effectively leads to improved skin tone and elasticity. Carboxytherapy can be used to reduce wrinkles that appear in the area of the lower and upper eyelids. There is also a repigmentation process due to an increase in the amount of oxygen in the tissue and intensification of blood flow, which leads to increased melanin production. There is a reconstruction of the eye area and a reduction of static wrinkles, dark circles and fat pads. Carboxytherapy also leads to improved microcirculation, improved vascular function, reduction and diminution of spider veins, which is beneficial for lower limb telangiectasias [62]. The effectiveness of carboxytherapy treatment in the area of cosmetic indications has been proven by numerous studies. The effects of carrying out this procedure are usually noticeable already at the initial stage of the conducted therapy [68].

*2.2. Non-Invasive Methods*

Non-invasive or minimally invasive methods are the second group of ways to correct the skin and its condition in a cosmetics cabinet. These include [69]:

- needle-free mesotherapy,
- application of ultrasound,
- oxygen infusion,
- chemical peels
- micro-needle mesotherapy.

### 2.2.1. Needle-Free Mesotherapy

Needle-free mesotherapy uses the phenomenon of stimulation of the skin by electrical impulses. The application of electrical potentials opens channels, allowing active substances to penetrate into the deeper layers of the skin. This mesotherapy is the least invasive skin stimulation available on the market. Using the phenomenon of electroporation, it is used to rejuvenate aged skin [70].

For this treatment, three phenomena occur simultaneously: electroporation, pinocytosis and sonosphere, or micromassage. Electroporation involves the use of a pulsed electromagnetic wave to destabilize the cell membrane, resulting in the formation of pores in the membrane [68]. Pinocytosis is the uptake of fluids and nutrients by skin cells through the convexity of the cell membrane and the tubules formed in it. The molecules of active substances contained in cosmetics also work after the treatment, which prolongs its effectiveness. The most commonly used ingredients for mesotherapy are organic acids, organic silica, hyaluronic acid, peptides, glutathione and vitamins. The treatment is very pleasant and painless, dedicated to clients who are afraid of more invasive procedures. Effects after this treatment include skin rejuvenation, brightening of spots, revitalization, hydration and skin firming. The treatment takes about 30 min, and is applied once a week in a series of 5–10 treatments, depending on the client's skin condition [71]. Needleless mesotherapy is used to prevent wrinkles, skin aging, to eliminate cellulite, reduce visible stretch marks, scars, and sagging skin. It is also used for skin firming, body contouring and to stimulate circulation [70].

### 2.2.2. Ultrasonic Waves

Ultrasound is sound waves with a frequency of 20,000 Hz. Their high frequency makes them inaudible to the human ear. It has long been a popular non-invasive method of

renewal and regeneration of aged skin among clients. Depending on the frequency setting of the power of such a wave, different depths of the skin can be affected. Ultrasound widens the intercellular space, which allows for better penetration of active substances contained in cosmetics used in the treatment. The deep skin massage performed by the device stimulates cells to regenerate. The treatment has a strong effect on fibroblasts, smoothing wrinkles and improving skin tone. It is called a 'lift without a scalpel' [72].

### 2.2.3. Oxygen Infusion

A new cosmetic treatment is oxygen infusion. This method involves the injection of various active substances into the deeper layers of the skin, and the carrier for these substances is pure oxygen injected into the skin under increased pressure. It is a non-invasive method by injecting active substances into the layers of the skin, without puncturing. As a result of such treatment, skin cells are oxygenated and their metabolism is increased, which increases their energy state and ability to regenerate. More efficient cell division and expansion of blood vessels (angiogenesis) are achieved, which is important in maintaining skin tone and firmness. By stimulating fibroblasts to produce collagen and elastin, the skin gains firmness and elasticity. The effects of oxygen infusion include skin refreshment and oxygenation, improved skin tone, increased tension and firmness, shallowing of wrinkles and the acceleration of healing and regeneration processes. It is primarily dedicated to clients with aged skin that requires immediate regeneration. The effects of the treatment are already visible after the first treatment, but appropriate home care is recommended after the treatment in order to consolidate the effect. Oxygen infusion treatment via highly concentrated oxygen molecules also exhibits strong antibacterial and anti-inflammatory effects, through which it is recommended for acne-fighting therapies [73].

### 2.2.4. Chemical Peeling

Chemical peeling is classified as a minimally invasive procedure. It involves the use of chemicals to remove lesions caused by skin aging or trauma. It is currently the most popular skin resurfacing treatment. The most common treatments are performed using combined surface peels. Surface peels separate the corneocytes from the layers of cells within the stratum corneum, even down to the stratum spinosum. Treatments with chemical acids achieve improvements in the dermis by stimulating fibroblasts, through which there is an increase in the production of collagen and glycosaminoglycans. Factors that affect the depth of penetration of superficial peels depend on the chemical compound used, the time of application, the client's skin type and the preparation of the skin before treatment. Chemical compounds used for this type of treatment include glycolic acid at a concentration of 20–70%, mandelic acid 50–70%, pyruvic acid 40–60%, salicylic acid 20–30%, trichloroacetic acid (TCA) 10–25%, retinoids and Jassner's solution. The frequency of treatments is from 10–30 days in a series of 6–8 treatments [74,75].

### 2.2.5. Microneedle Mesotherapy

Micro-needle mesotherapy is a treatment that stimulates the epidermis and increases its permeability, allowing highly concentrated ingredients of the right molecular size to penetrate deeper and at a higher concentration. Mechanical stimulation stimulates the skin's natural regenerative abilities. Micropunctures made with sterile, disposable needles create several thousand microchannels. Perforation of the epidermis is carried out in a strictly controlled manner—both its frequency and depth of penetration are adjusted according to the client's feelings [76]. During puncturing, blood vessels are damaged, so that a small amount of blood is released and cytokines are secreted into the extravascular space, affecting skin regeneration processes. This results in the release of so-called growth factors, which include [77]:

- transforming growth factor alpha (TGF-α),
- transforming growth factor beta (TGF-β),

- fibroblast growth factor 2 (FGF-2) and basic fibroblast growth factor (bFGF); these strongly stimulate fibroblasts to produce proteins that form the extracellular matrix (ECM),
- platelet derived growth factor (PDGF),
- epidermal/epidermal growth factor (EGF),
- connective tissue growth factor (CTGF).

The treatment achieves a lifting effect, improves firmness and elasticity and stimulates collagen and elastin production. The frequency and number of treatments are adjusted to the client's skin needs. The best results are obtained after a series of 3–4 treatments performed at intervals of 3–4 weeks.

## 3. Active Compounds of Cosmetics for Aged Skin

The active ingredients present in dermocosmetics for aged skin care are very diverse, which is related to the numerous theories of skin aging. Their function is to restore a more youthful appearance and delay and repair skin damage. These substances include cell stimulators, estrogen-type anti-aging agents, free radical scavengers and sunscreens, as well as skin exfoliating, strengthening and tightening agents.

### 3.1. Phytoestrogens

An ingredient of dermocosmetics intended for women of mature age, especially menopausal women, are phytoestrogens, which are a cosmetic discovery of the 20th century. These are plant-derived compounds that show structural similarity to estrogen [78]. The ability to bind to the ER (both ERα and ERβ) is also important in anti-aging cosmetics. ERβ receptors in the skin are found on keratinocytes, melanocytes, dendritic cells and vascular endothelial cells. Their role is related to influencing keratinocyte proliferation and differentiation, facilitating proper keratinocyte adhesion, formation of keratohyalin deposits, inhibiting IL-12 and TNF-α formation and regulating melanin secretion. ERα receptors are found on fibroblasts and macrophages. They are involved in the stimulation of type I and II collagen production by affecting the secretion of transforming growth factor (TGF-β1) and basic fibroblast growth factor (bFGF) [78]. The potency of these compounds is weaker than that of 17-β-estradiol. The most important and best known class of phytoestrogens are isoflavones (genistein, daidzein). Other groups of compounds belonging to the phytoestrogens include lignans, stilbenes, coumestans, coumarins, as well as dihydroxychalcones and triterpenoids. Among the isoflavones with the strongest estrogen-like effects are genistein and daidzein [79], as well as resveratrol [80], a polyphenolic compound belonging to the stilbene group. Phytoestrogens have several biological mechanisms of action, including antioxidant activity [81,82], reduction of UV radiation-induced skin damage [83] reduce melanogenesis [84] and improve skin vascularization in postmenopausal women [85]. These biological properties have found applications in anti-aging skin care. Studies conducted in vivo have shown the effectiveness of phytoestrogens against skin aging [86–89].

### 3.1.1. Isoflavones (Genistein, Daidzein)

The sources of obtaining isoflavones such as genistein, genistin and daidzenin, which are phytoestrogens, are mainly plants of the Leguminosae family, e.g., *Glycine max* (L.) Merr. [90], *Pueraria candollei* Grah. ex Benth. var. *mirifica* [91], *Styphnolobium japonicum* (L.) Schott (syn. *Sophora japonica* L.) [92] and *Trifolium pretense* L. [93]. In 2018, Savoia et al. [94] demonstrated in their study the protective effect of genistein against peroxidative damage in fibroblasts and keratinocytes by modulating the release of nitric oxide (NO) and reactive oxygen species (ROS). Genistein also increased the glutathione (GSH) content and improved mitochondrial function [94]. Genistein inhibits UVB-induced oncoprotein expression and reduces the risk of sunburn and tumorigenesis. The efficacy of phytoestrogens is enhanced by the appropriate chemical form, the use of liposomal carriers and the simultaneous use of other ingredients with anti-aging effects [95]. From a safety assessment based on the available aglycone data form of genistein and daidzein, and taking into account the

potential endocrine effects, the Scientific Committee on Consumer Safety (SCCS) believes that the use of genistein and daidzein in cosmetics up to a maximum concentration of 0.007% and 0.02%, respectively, is safe [96].

### 3.1.2. Resveratrol

Resveratrol (3,5,4-trihydroxystilbene) is a polyphenolic phytoelaxin synthesized in some plants in response to stress factors and fungal infections [97,98]. It occurs naturally in high concentrations in red grapes, berries (cranberries, black currant, strawberries, raspberries) and peanuts. It is a potent antioxidant, modulator of genetic expression and inhibitor of inflammatory mediators. It is a potent inhibitor of NADPH, lipid peroxidation and an effective free radical scavenger, 17 times more potent compared to the Coenzyme Q10 analogue idebonone [97,99]. Resveratrol affects the mechanism of skin aging. It inhibits damage to the genetic material affecting the proper work of enzymes of the sirtuin class. Sirtuins (SIRT 1 and SIRT2) are enzyme proteins that normalize a cell's stress resistance and repair capacity and prolong cell life. The activity of sirtuins decreases with age [100]. Resveratrol is one of the most effective activators of these enzymes. By increasing their efficiency, resveratrol improves the skin's natural regenerative processes, as well as enhances the cells' resistance to dangerous external factors, including UV radiation. Resveratrol can also protect the skin from photoaging, reduce lipid peroxidation and reduce leukocyte infiltration [101]. In human keratinocytes, resveratrol blocked activation of the transcription factor kappa B (NF-κB) pathway, a protein responsible for intracellular signaling in the body's defense pathomechanisms and inflammatory processes induced by TNF-$\alpha$, IL-1, IL-6 and IL-8 [102–104]. Sticozzi et al. [105] also showed that resveratrol in vitro protects human keratinocytes from tobacco smoke. The anti-aging effect of resveratrol was proven in a study involving 55 women aged 40–60 years. When using a night cream containing resveratrol (1%), baicalein (0.5%) and vitamin E (1%) for 12 weeks, a marked improvement in skin firmness and elasticity, smoothing of fine wrinkles and reduction of hyperpigmentation was observed [106]. Resveratrol, as an anti-aging substance, has an inhibitory effect on the glycation of supporting proteins in the skin. Glycation involves the destruction of collagen and elastin by sugars. With age, the level of glycation processes in the body increases. There is a stiffening of collagen and elastin fibers. Their regenerative capacity decreases, resulting in increased inflammation and slower healing; at the same time, the skin takes on a yellowish color. Research on the effectiveness of resveratrol as a comprehensive anti-aging treatment was conducted by Brinke et al. [107]. The purpose of this study was to evaluate the topical effects of an emulsion containing 2% resveratrol on age-related skin lesions. The product was applied once a day for 8 weeks, and then selected skin parameters were evaluated. The results confirmed the anti-aging nature of the resveratrol emulsion. Indeed, after 8 weeks of regular use, the level of skin elasticity (+5.3%), skin density (+10.7%) increased, skin roughness (−6.4%) and dispensability (−45.9%) decreased, and the intensity of skin redness decreased [107]. The indication of the study confirms the anti-aging nature of emulsions containing resveratrol.

### 3.1.3. Stem Cell Extracts

With aging, the number of cells in tissues and organs with slowed regenerative processes increases. Skin stem cells are responsible for regenerating damaged or worn out basal layer cells and fibroblasts [108]. High hopes are placed on stem cells in both regenerative medicine and anti-aging cosmetology [109]. Extracts from plant cell cultures containing a mixture of active ingredients (primary and secondary metabolites) can be used in minimal concentrations in the formulation of cosmetics [110]. Clinical studies have proven that plant stem cell extracts used in cosmetics are readily absorbed by the epidermis, causing an almost immediate renewal of skin cells. Cosmetic companies are using plant stem cells in the production of anti-aging cosmetics [110–116]. Plant stem cell extracts are responsible for many positive cosmetic effects, such as [110,116]: extending the life of fibroblasts and stimulating their activity (e.g., *Oryza sativa* L., *Gardenia jasminoides* J.Ellis); increasing the

flexibility of the epidermis (e.g., *Symphytum officinale* L., *Capsicum annuum* L., *Opuntia* spp.); regulating cell division (e.g., *Oryza sativa* L., *Lotus corniculatus* L.); rebuilding damaged epidermis (e.g., *Panax ginseng* C.A.Mey., *Opuntia* spp.); activating DNA repair of the cells, protecting them from oxidative stress (e.g., *Rubus ideaus* L., *Lycopersicon esculentum* Mill., *Citrus limon* (L.) Osbeck); and protecting against UV radiation (e.g., *Vigna unguiculata* (L.) Walp., *Opuntia ficus-indica* (L.) Mill.). The first anti-aging plant stem cell extracts from *Malus domestica* were introduced in 2008 [117]. Liposomes were used as carriers for the extracts. An experiment was conducted using human fibroblasts that were exposed to $H_2O_2$. After two hours of exposure, the skin fibroblasts had typical signs of aging. Subsequently, one part of the cells was placed in a 2% apple (*Malus domestica* stem cell) stem cell extract, while the other, a control sample, was placed in a neutral environment. The results of the experiment showed that the apple stem cell extract reversed the aging process of the fibroblasts and led to the stimulation of the expression of the antioxidant enzyme hemeoxygenase-1, while in the control sample the fibroblasts were irreversibly damaged [117]. When used as an ingredient in cosmetics, apple stem cell extract shows great potential for reducing wrinkles in the eye area, the so-called 'crow's feet.' Using an eye cream with the extract reduced the depth of wrinkles after 2 weeks by 8%, and after 4 weeks by as much as 15% [118]. Polish scientists from Wroclaw have isolated stem cells extracted from red deer antlers (MIC-1). They are pluripotent cells and can differentiate into all cells that make up the antler: osteocytes, chondrocytes, neurons, fibroblasts and keratinocytes. Using stem cell culture from deer antlers, a new complex for cosmetic use was developed. In addition to MIC-1 cells, the complex contains unsaturated fatty acids, phospholipids and collagen peptide active factors, mineral salts and sugars in its composition. The complex contains growth factors that stimulate fibroblasts and specific epidermal stem cells. This leads to natural skin renewal [119]. After the application of cosmetics containing active substances from deer antlers, the density, firmness and elasticity of the skin increase and wrinkles become shallower. The process of bio-renewal of the dermis and basal layer that occurs after using the products contributes to skin rejuvenation [120]. Stem cell extracts are very good ingredients to cosmetics designed for aged skin care.

### 3.1.4. Vitamins

Vitamins play a very important role in slowing down the aging process, especially those with strong antioxidant potential such as vitamin A and its derivatives—retinoids, vitamin C, vitamin E and coenzyme Q10. Cosmetics containing vitamins in their composition have a multidirectional effect. They not only inhibit the formation of oxidative stress in the skin, but also protect against the adverse effects of external factors, the so-called extrinsic aging. Cosmetics rich in antioxidant vitamins prevent inflammatory processes underlying the aging process [121].

### 3.1.5. Vitamin A and Its Derivatives—Retinoids

Retinoids are among one of the most effective and widely used ingredients in anti-aging cosmetics. They exhibit a lipophilic nature, so they penetrate the epidermal barrier very easily, passing into the deeper layers of the skin. Vitamin A derivatives have a smoothing effect on the skin, as well as making it more elastic. It eliminates fine wrinkles, reduces hyperpigmentation and increases the production of collagen fibers; the number and activity of fibroblasts reduces the activity of metalloproteinases and also improves skin angiogenesis [122]. Retinoids stimulate the synthesis of collagen I, III, VII, as well as procollagen I and fibrillin. They promote collagen remodeling, and stimulate the proliferation and differentiation of epidermal cells, resulting in an increase in the cohesiveness of the stratum corneum and thickening of the granular layer. They inhibit tyrosinase and limit the transfer of melanosomes, increasing the exfoliation of melanocytes so that hyperpigmentation becomes less visible. The complex mechanism of action of retinoids makes deep wrinkles shallower, skin radiant, well-hydrated, smoother, more elastic and smoother to the touch [123]. The most common forms of vitamin A in cosmetics include retinol and its

esters, palmitate and retinyl acetate, and β-carotene [124]. Due to safety reasons (risk of teratogenicity), the Scientific Committee on Consumer Safety, Secretariat at the European Commission, Directorate General for Health and Food Safety recommends a maximum retinoid concentration of 0.05% retinol equivalents (RE) in body lotions and 0.3% RE in hand and face creams, as well as other leave-on or rinse-off products for cosmetics in the EU [124]. Rakuša and Roškar [121] studied 35 cosmetic creams containing various retinol derivatives. The content of the active ingredient per retinol ranged from 160 ng/g–19 mg/g. The application of cosmetic formulations containing retinol or its derivatives is one of the most effective methods of preventing skin aging [123].

### 3.1.6. Vitamin C

Vitamin C and its modified molecules protect the skin from oxidative damage, rejuvenate photoaging skin, brighten skin, have anti-inflammatory effects and reduce erythema after sun exposure or laser treatments. The anti-aging effect of vitamin C is due to the fact that it is essential as a cofactor for the enzymes lysyl hydroxylase and prolyl hydroxylase, which are essential in the biosynthesis of collagen (types I and III). Thus, by stimulating these stages of biosynthesis, ascorbic acid will increase collagen production, leading to wrinkle reduction [125]. Topical application of vitamin C increases levels of tissue inhibitors of collagen-degrading matrix metalloproteinase 1 (MMP-1) [126]. Garre et al. [127] reported that the application of a cosmetic formulation containing ascorbic acid to skin sections in vitro protected against oxidative damage and photoaging-induced protein loss. In tests on study participants, it provided very good skin hydration, and brightened and smoothed the skin [128]. The beneficial properties of vitamin C make it a popular choice for use in anti-aging and regenerative cosmetics [111]. Topical application of vitamin C is the cornerstone of anti-aging management [88]. There are several forms of vitamin C used in cosmetology. The first, non-permanent form is the active form of vitamin C, L-ascorbic acid. To increase the durability of cosmetics containing ascorbic acid, its esterified derivatives are used: ascorbyl tetraisopalmitate, ascorbyl palmitate-6-ascorbyl palmitate, ascorbyl phosphate magnesium and sodium salt [128–130]. Exposure to UV rays reduces vitamin C content in the skin by about 70%. External application of this vitamin protects the skin from the dangerous effects of UV radiation, especially when using vitamin C, vitamin E and ferulic acid simultaneously [129]. Vitamin C stimulates ceramide synthesis and increases the amount of TIMPs-tissue inhibitors that inhibit MMP1 metalloproteinase [129]. A study of 19 people aged 36–72 with moderately UV-damaged facial skin showed that regular use, over a 3-month period, of a preparation containing 10% ascorbic acid improves the overall appearance of the skin, reduces roughness and minor skin defects, shallows wrinkles, increases flexibility of the skin and unifies skin tone by reducing significant skin yellowing, compared to a control group that used the base alone for the same period of time [129]. In the care of aged skin, L-ascorbic acid derivatives perform better, among which lipophilic ascorbyl tetraisopalmitate is the most durable. It can be used in formulations in combination with retinol and ferulic acid, thus significantly increasing the anti-aging and brightening properties of the compound. Unlike ascorbic acid, it can be used for sensitive skin [128].

### 3.1.7. Vitamin E

Vitamin E is considered the most important lipophilic exogenous antioxidant. Vitamin E is used in cosmetics as an active ingredient, with occlusive and emollient action [131]. It is also stabilizer of other, non-permanent components of a cosmetic product [106,132]. It builds into cell membranes, prevents oxidative stress-induced peroxidation of cell membrane lipids and reduces UV-induced erythema and swelling [106]. It also exhibits strong anti-aging properties [133,134]. According to studies, α-tocopherol has better antioxidant properties compared to tocopherol esters, which require prior hydrolysis in the skin [121]. In their research, Rakuša and Roškar [121] determined the concentration of tocopherol in 49 cosmetic formulations. Tocopherol was present in them at a concentration of 8.5 μg/g to

16 mg/g. It has also been shown that when using a formulation with tocopheryl acetate every day for five consecutive days, it is possible to fully inhibit erythematous lesions resulting from the influence of UV radiation [128]. In addition, vitamin E, present in cosmetics, supports the reconstruction of damaged tissue, increases blood supply to the skin, improves the elasticity of the connective tissue and accelerates the processes of collagen and elastin biosynthesis occurring in the dermis [128].

### 3.1.8. Coenzyme Q10, Alpha-Lipoic Acid and Idebenone

A natural antioxidant that is a quasi-vitamin is ubiquinone, otherwise known as coenzyme Q10 [135]. The content of coenzyme Q10 in the skin is relatively low. There is 10 times less ubiquinone in the dermis than in the epidermis, so the epidermis can benefit from topically applied ubiquinone [136]. Coenzyme Q10 also acts as an antioxidant in the skin. It protects glutathione from its breakdown, slows DNA damage in keratinocytes, increases glycosaminoglycans and protects collagen from breakdown. Ubiquinone minimizes the symptoms of extrinsic and intrinsic aging [128]. Coenzyme Q10 in the form of ubiquinone is a popular ingredient in anti-aging cosmetics, among others, in which it is usually found in concentrations of $\leq 0.05\%$ [137]. An analog of coenzyme Q10, idebenone, is also used in cosmetics. It exhibits antioxidant activity comparable to lipoic acid and stronger than ubiquinone [128]. Creams with alpha-lipoic acid smooth fine wrinkles, reduce photoaging lesions, reduce the intensity of hyperpigmentation and reduce inflammatory reactions [128].

### 3.1.9. Plant Extracts

Plant extracts are very good for daily skin care, and because of their wide range of applications, plant extracts are called one of the multifunctional cosmetic ingredients [138–140].

### 3.1.10. Extract of *Scutellaria baicalensis* Georgi (Lamiaceae)

*Scutellaria baicalensis* extract is widely used in cosmetics dedicated to aged skin. It can also be used as a cosmetic ingredient for whitening, sun protection and anti-aging [141,142]. Following the discovery of Hayflick's theory, scientists sought to find a way to activate telomerase in somatic cells. They isolated a natural substance found in the roots of *S. baicalensis*, baicalin, which increases telomerase expression [143]. The activation of telomerase by baicalin has only been discovered in recent years. In vitro studies proved that aging fibroblasts housed for 2 months in a baicalin environment restarted their divisions. This was due to the elongation of the DNA strand that forms the telomere [143]. Using baicalin extracted from *S. baicalensis*, the anti-aging raw material was developed to prevent cellular aging by stimulating telomerase expression in fibroblasts. The effect of baicalin on a 50-year-old person would be to rejuvenate her skin fibroblast population by 10 years [142]. Baicalin has been shown to delay fibroblast aging by increasing the number of cell divisions by 10%, assuming that the total number of fibroblast divisions is 50. According to the above experiments, it was noted that the test substance restores fibroblast activity to a state comparable to skin ten years younger [141]. In in vivo tests, the positive effect of baicalein on the skin was confirmed. The study was conducted in a group of 20 women aged 35–45 years. After two months of using a cream with baicalin-containing raw material, an increase in skin elasticity and firmness of about 12.5% was noted. An improvement in the skin microsculpture and 13% reduction in wrinkles was observed [142]. The raw material has become a very promising product in anti-aging medicine. This modern and advanced extract can be successfully used in anti-aging and rejuvenating formulas. It has a beneficial effect on the skin, improving its appearance, as well as its condition, while delaying the aging process [142].

### 3.1.11. *Centella asiatica* Extract

*Centella asiatica* L. Urb. (Apiaceae) extract is used in cosmetics for aged skin due to its valuable anti-aging, anti-inflammatory, anti-bacterial and collagen synthesis stimulating properties [139,140]. Active substances belonging to the group of bioflavonoids (quercetin

and kaempferol), phytosterols, triterpene saponins (madecasoside and asiaticoside), among others, have been isolated from the plant [139]. Madecasoside soothes and reduces the allergic reaction in the skin caused by external factors. In addition, it stabilizes collagen fibers and increases cell proliferation. Asiaticoside stimulates cell granulation, accelerates regeneration and scarring processes, inhibits hyaluronidase and stimulates the synthesis of glycosaminoglycans in the skin [139]. *C. asiatica* extract activates fibroblasts to biosynthesize collagen (mainly type I collagen) and elastin. Due to its effects, it is widely used in anti-aging cosmetics. When applied to the skin, *C. asiatica* extract reduces wrinkles. It also exhibits antioxidant and antioxidant activity and accelerates angiogenesis. Asiatic acid present in the plant extract increases the synthesis of collagen and proteoglycans increases wound healing. The valuable moisturizing properties of *C. asiatica* are confirmed by numerous scientific studies [139,140].

### 3.1.12. Bakuchiol

Bakuchiol is a plant-derived substance that is used by cosmetics manufacturers as an ingredient of anti-aging preparations. It is a meroterpene phenol found mainly in the seeds of the Indian plant *Psoralea corylifolia* [144,145]. Research results published by Chaudhuri and Bojanowski [146] showed that retinol and bakuchiol have a similar gene expression profile, especially in key genes and proteins which act as countermeasure for aging. Bakuchiol, unlike retinol, has an excellent photochemical and hydrolytic stability, and a good safety profile. It can be used during the day due to its photostability [146]. Studies conducted by Chaudhuri and Bojanowski [146] have shown that the regular use of a cream containing 0.5% bakuchiol for 12 weeks significantly improves the smoothness of the skin and reduces the appearance of fine wrinkles. In addition, preparations with bakuchiol brighten the skin and reduce sun discoloration [147]. It can be used even in people with very sensitive skin [148].

### 3.1.13. Peptides

Peptides are very popular cosmetic ingredients. Glutathione (γ-glutamyl-cysteyl-glycine) is a GSH tripeptide which is one of the first peptides used in cosmetology. It contains the amino acid cysteine -SH, which gives this molecule its antioxidant activity. Glutathione levels in the body decline markedly with age, which contributes to skin aging [149]. Glutathione is also helpful in eliminating skin discoloration of various etiologies [149]. Carnosine (β-alanyl-l-histidine) scavenges reactive oxygen species (ROS) formed by the peroxidation of cell membrane fatty acids during oxidative stress. Carnosine has also been shown to prevent the effects of protein glycation, which leads to damage and fragility of collagen fibers [150]. Another popular stimulating peptide belonging to the matrikines family that is often used in cosmetics is (a fragment of the type I procollagen sequence of) the Pal-KTTKS pentapeptide. It is found in combination with palmitic acid, to which it owes its good penetration through the skin; the main function of this peptide is to increase the production of type I and II collagen and fibronectin [151]. Tetrapeptide Arg-Gly-Asp-Ser (RGDS) is the structure of fibronectin responsible for the ability of this protein to bind to collagen and cell membranes, thus accelerating wound healing. The cyclic peptide RGD, on the other hand, binds to integrin receptors, thus showing anti-wound healing properties [151]. The Pal-Gly-Gln-Pro-Arg peptide (Pal-GQPR) is a fragment of a natural circulating IgG protein which contributes to the reduction of UV-induced IL-6 release in keratinocytes and fibroblasts, leading to improved skin firmness [152]. Peptides in cosmetology represent a novelty in the prevention of skin aging. They are distinguished by their ability to penetrate the epidermis and reach the deep layers of the dermis. Peptides are involved in many skin processes: stimulation of protein synthesis, melanin production, migration of pro-inflammatory cells, modulation of fibroblast proliferation, as well as formation of capillaries. The effects of the peptides are visible only after 8–12 weeks of regular use [152]. It manifests itself by increasing the firmness and elasticity of the skin, strengthening the blood vessels present in the skin, reducing irritation and reducing the

appearance of facial wrinkles. In addition, the peptides stimulate mechanisms that protect the skin from harmful UV radiation and also prevent the formation of facial wrinkles at an early age [152].

## 4. Summary

The skin is the largest human organ, and the signs of aging, a natural biological process, are clearly visible on it and become more pronounced with age. The result of these processes are wrinkles, and the main causes of wrinkles are the loss of collagen, loss of hyaluronic acid and weakening of the hydrolipid barrier of the epidermis. The dermis, thanks to its high content of collagen and hyaluronic acid, maintains its cohesion, density, elasticity, flexibility and firmness, which is what determines its youthful appearance. Additionally, skin is exposed to external factors such as UV radiation, smog (fine suspended particulate matter, PM 2.5), and free radicals on a daily basis, which can accelerate the aging process. The main problem with aged skin is the progressive atrophy of the skin, associated with degenerative processes in all its layers. The loss of body fat on the face makes it look tired and sunken. Its susceptibility to mechanical deformation increases, and hyperpigmentation intensifies. To solve these problems, both invasive and non-invasive methods of skin treatments are available. Invasive methods performed by a dermatologist, such as PRP treatment, needle mesotherapy or botulinum toxin injection, may provide faster results, but are associated with a greater risk of complications and allergic reactions. On the other hand, non-invasive methods, such as needle-free mesotherapy, ultrasound, oxygen infusion, chemical peels and micro-needle mesotherapy are less risky, but may take longer to see results. Ultimately, the best approach for the patient will depend on their individual problems and skin preferences. In addition to salon-based care for aged skin, it is also important to have a properly selected cosmetic daily routine. Cosmetics used on a daily basis should provide appropriate moisture, UV protection, combat free radicals and act regeneratively. Ingredients that prevent the negative effects of oxidative stress on skin are vitamin C, vitamin E and coenzyme Q10. Baicalin, obtained from the Baikal skullcap (*Scutellaria baicalensis*), and retinol or its plant-based counterpart bakuchiol have very good anti-aging properties. Regular and long-term use of these ingredients allows the improvement of the appearance of the skin, reducing the visibility of fine wrinkles and the negative effects of free radicals on the skin.

**Author Contributions:** Conceptualization, E.R. and K.W.; writing—original draft preparation, E.R. and K.W.; writing—review and editing, E.R., K.W., E.P. and K.D.S.S.; supervision, K.D.S.S. and E.P. All authors have read and agreed to the published version of the manuscript.

**Funding:** This research received no external funding.

**Institutional Review Board Statement:** Not applicable.

**Informed Consent Statement:** Not applicable.

**Data Availability Statement:** Data available on request.

**Conflicts of Interest:** The authors declare no conflict of interest.

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
