# Peer review of "Dermatological Management of Aged Skin"

_cosmetics, doi:10.3390/cosmetics10020055_

Round 1
Reviewer 1 Report (Previous Reviewer 1)
The skin from the face is also known as hairy skin since we have pilossebaceous units in the face. Glabrous skin is the skin from palm, feet, lips etc
The differences regarding glabrous skin x hairy skin is related with differences in the SC thickness, presence - or not - of pilossebaceous units and also the perception. Please improve the lines from 112-115 with this information. References below:
Gould, J. (2018). Superpowered skin. Nature, 563(7732), S84-S85. The authors have pointed: "There is consent of patients for the publication of photos.The treatments were performed in an aesthetic medicine office as a standard beauty treatment. They were not a medical experiment."
This is an ethical situation. You, as the professional which are performing procedures, even minimally invasive, are in an unequal relation with your patients. The consent to publish photos without an ethical committee continue to be an ethical problem. You CAN NOT provide images or informations from patients without an ethical committee which have approved your work prior the start. Even if they gave their consent. One point is to give the consent after an ethical committee approval, other point is to ask without this concern.
Other point: even studies with cosmetics, topical application without any invasive procedure, it is necessary to have a medical doctor following the study. For invasive situations such as you have pointed is mandatory. It is important not to trivialize asthetics procedures since some of them are invasive and can have side effects.
One point is to make a review from your field following the literature, other point is to confund the review with your personal work. This should be known for future studies and publications.
This can be hard and annoying to read but consider that other reviewers could just reject your paper without any consideration since some ethical problems could be detected.
Author Response
Please see the attachment.

Reviewer 2 Report (Previous Reviewer 2)
Dear Authors,
thank you for resubmitting your manuscript, now in it’s revised form. I still have some minor suggestions, that could improve the quality of the manuscript:
Please make Key words more specific- indeed you have done modifications, but they are still not very suggestive/or repetitive.
Lines 1064-1065: "The loss of fat pads causes the face to appear tired and sunken"- please rewrite using more scientific/specific terms.
Lines 1065-1066: "age spots"- hyperpigmentation
Lines 1067-1068: "cosmetic doctor"- do you refer to a dermatologist?
Lines 1108/1114-"selected home care routine"- selected cosmetic daily routine
Please carefully revise English synthax and grammer.
Also please recheck References according to Cosmetics Instructions for authors.
Kind regards,
the Reviewer.
Author Response
Please see the attachment.

Reviewer 3 Report (Previous Reviewer 3)
Dear Authors,
All comments and suggestions are included in the text.

Round 2
Reviewer 1 Report (Previous Reviewer 1)
After the reviews, the present work is fine. Please do not forget to submit your research for an ethical committee prior the study and publication.
Author Response
Dear Reviewer,
We are thankful to the reviewer for reviewing again our manuscript. Ultimately, we decided not to include any photos in our work.
Yours sincerely,
Authors
Reviewer 3 Report (Previous Reviewer 3)
Minimal comments and suggestions for the authors are included in the text.

Author Response
Dear Reviewer,
We are thankful to the Reviewer for reviewing again our manuscript.
All changes in the text you are suggested, are highlighted in yellow.
Yours sincerely,
Authors
This manuscript is a resubmission of an earlier submission. The following is a list of the peer review reports and author responses from that submission.
Round 1
Reviewer 1 Report
Introduction:
Very nice review about skin, however I could suggest that you also include the differences between hairy and glabrous skin since the following paper seems to be more focused in the hairy skin.
line 60 - please include the term "stratum corneum"
lines 75-78: which substances regulate the moisture in dermis? Elastin? Please clarify. It should be also important to include that the dermis is about 60-70% water instead of 45-60% in the SC. Besides, the SC presents a lot of hygroscopic substances.
lines 95-96: is this right? Wrinkles in men seems to be more pronounced. Is the reference number 7 responsible for all the information in this paragraph? I advice to reparagraph with more literature. We already know that even female skin can present high sebum levels. There is also a dependece of weather, for example. Maia Campos, P. M., Melo, M. O., & Mercurio, D. G. (2019). Use of advanced imaging techniques for the characterization of oily skin. Frontiers in Physiology, 10, 254. De Melo, M. O., & Maia Campos, P. M. B. G. (2018). Characterization of oily mature skin by biophysical and skin imaging techniques. Skin Research and Technology, 24(3), 386-395. Jo, D. J., Shin, J. Y., &Na, S. J. (2022). Evaluation of changes for sebum, skin pore, texture, and redness before and after sleep in oily and nonoily skin. Skin Research and Technology, 28(6), 851-855.
For this paragraph I also could suggest the influences of culture in the skin care habits. For example, korean men seems to be more open for skin care than other cultures. Also, this paragraph gaves the idea that men did not suffer too much sun damages, what is not correct. Ficheux, A. S., Gomez-Berrada, M. P., Roudot, A. C., & Ferret, P. J. (2019). Consumption and exposure to finished cosmetic products: A systematic review. Food and Chemical Toxicology, 124, 280-299. Infante, V. H. P., Bagatin, E., & Maia Campos, P. M. (2021). Skin photoaging in young men: A clinical study by skin imaging techniques. International Journal of Cosmetic Science, 43(3), 341-351. McKenzie, C.,Rademaker, A. W., & Kundu, R. V. (2019). Masculine norms and sunscreen use among adult men in the United States: A cross-sectional study. Journal of the American Academy of Dermatology, 81(1), 243-244.
Lines 145-146: I suggest to include one more line explaining that for phototypes III and IV are also evidences that photoaging can cause aging besides these skin types present a higher predispotion to have melanocitic alterations such as melasm or lentigo.
Passeron, T., & Picardo, M. (2018). Melasma, a photoaging disorder. Pigment cell & melanoma research, 31(4), 461-465. Yang, J., Zeng, J., & Lu, J. (2022). Mechanisms of ultraviolet‐induced melasma formation: A review. The Journal of Dermatology.
lines 135-147: please include that photoaging is a process, not just the clinical signals, what means that structural alterations can be observed prior the clinical signals, being important to improve the photoprotection habit among the population Kutlu Haytoglu, N. S., Gurel, M. S., Erdemir, A., Falay, T., Dolgun, A., & Haytoglu, T. G. (2014). Assessment of skin photoaging with reflectance confocal microscopy. Skin Research and Technology, 20(3), 363-372. Longo,C., Casari, A., Beretti, F., Cesinaro, A. M., & Pellacani, G. (2013). Skin aging: in vivo microscopic assessment of epidermal and dermal changes by means of confocal microscopy. Journal of the American Academy of Dermatology, 68(3), e73-e82.
lines 158-173: it is already known that visible light provenient from eletronic devices DID NOT present enough energy/dose to be responsible for a significant increase in the photoaging process by ROS/LOS.
This paragraph should be completely deleted and reconsidered. The effects of reference 23 are not comparable with other researchers already described since this is a study in fibroblasts, cells from dermis. Mann, T., Eggers, K., Rippke, F., Tesch, M., Buerger, A., Darvin, M. E., ... & Kolbe, L. (2020). High‐energy visible light at ambient doses and intensities induces oxidative stress of skin—Protective effects of the antioxidant and Nrf2 inducer Licochalcone A in vitro and in vivo. Photodermatology, photoimmunology & photomedicine, 36(2), 135-144.
Tsuchida, K., & Sakiyama, N. (2022). Blue light-induced lipidoxidation and the antioxidant property of hypotaurine: evaluation via measuring ultraweak photon emission. Photochemical & Photobiological Sciences, 1-12.
line 332: a before/after photo without any result do not sound scientifically reasonable to be published. The lights are different, the hair line is in a different perspective, please remove this photo.
line 359: same consideration as figure 1.
line 378: same consideration above.
line 448: same consideration above.
line 526: same consideration above.
If you include photography from participants an ethical committee should aprove the present study, otherwise ethical problems can be observed. Stem cells are not utilized direct in cosmetic formulations since all formulations present a quality control that do not allow the presence of these cells in a cosmetic cream. Mostly, they present metabolites from these cells. Besides, why the authors had include a commercial name in the middle of a scientifical review? Besides, steam cells are not so often in the scientifical literature for cosmetic products. The authors spent more than half page about it and did not mention new ingredients such as bakuchiol, products from algae or some essential oils.
Vitamin C Do the authors believe that vitamin C, a hydrophilic molecule, is able to penetrate until the dermis and improve the skin condition or its effects are more topical? The authors did not include a section to talk about topic/injectable/delivered substances...this should be considered.
Reviewer 2 Report
The manuscript represents an interesting essay describing invasive and non-invasive procedures for mature skin.
I permit to give some suggestions, which could enhance the quality of the manuscript:
Abstract- The authors should add the originality of the work. Also, the abstract is very general.
Line 11: “people” and entire manuscript- I would suggest patient/consumer
Line 15: “the types of treatments”- treatment types?
Please, make the Key words more specific.
Line 259: “antipollution cosmetics”- even the term exists, I would suggest another cosmetic claim, or please rewrite this sentence
Please, make the Key words more specific.
2.1. Invasive methods- in the Abstract of the manuscript you mention procedures which are performed in professional beauty salons- invasive methods are performed by dermatologists!
Line 280: “flabbiness”- do you mean skin laxity?
Line 333: please add reference to Fig. 1. Please add reference to Fig. 3-7 even there are your personal efficacy studies and results, together with the ethical approval (Ethics Committee)- protocol code and date of approval!
Your manuscript is entitled “Dermatological Management of Mature Skin”- as invasive methods and study examples you also refer to androgenetic alopecia, gynoid lipodystrophy (cellulite) …
2.2.4. Chemical Peeling – please describe in more detail keratolytic agents which are used- concentration, cosmetic active ingredients, cosmetic products and procedures assisted by dermatologists or performed in beauty salons and make a clear distinction of these category of treatments/cosmetic use.
Line 626: “multi-directional effects” – do you refer to multifunctional ingredients/cosmetics?
Line 866: not only plant extracts are “multifunctional cosmetics ingredients”, also other cosmetic/active ingredients have multifunctional effect.
Line 869: I would not use “anti-allergic and anti-inflammatory” effect for cosmetics.
Line 912: “Peptides are among the most growing objects …”- peptides are indeed innovative cosmetic ingredients- please reformulate!
Line 947: “skin discoloration”- do you refer to hypo-/hyperpigmentation?
Lines 948-949: invasive methods are not performed in beauty salons, eventually by dermatologists!
Lines 942-958: Summary- the authors should avoid to repeat ideas from the manuscript, which personally, I find unnecessary. The authors should focus on the topic from their own findings.
Please revise References according to Cosmetics Instructions for authors.
Kindly, could you recheck the English grammar and the syntax of the manuscript?
Kind regards!
Reviewer 3 Report
Dear Sirs,
All suggestions and comments for the authors are included in the text. Authors should pay special attention to citing references both in the text and in the list of references.
